# No evidence for adult smartphone use affecting attribution of communicative intention in toddlers: Online imitation study using the Sock Ball Task

**Solveig Flatebø**[ID], **Gabriella Óturai**[ID], **Mikołaj Hernik**[ID]*

Department of Psychology, UiT The Arctic University of Norway, Tromsø, Norway

* mikolaj.l.hernik@uit.no

**Data Availability Statement:** All data and stimuli are available from the OSF database (doi.org/10.

## Abstract

Adults infer others' communicative intentions, or lack thereof, from various types of information. Young children may be initially limited to attributions based on a small set of ostensive signals. It is unknown when richer pragmatic inferences about communicative intentions emerge in development. We sought novel type of evidence for such inferences in 17-to-19-month-olds. We hypothesized that toddlers recognize adults' smartphone use in face-to-face interactions as incongruous with ostension and would rely on this interpretation when inferring the communicative intention of a model in a new imitation task conducted entirely online, dubbed the Sock Ball Task. In Experiment 1 with a between-subject design, we tested the hypothesis by assessing toddlers' ($N = 48$) imitation of sub-efficient means and the goal-outcome presented by a model, who interrupted her ostensive demonstration either by using a smartphone or by fiddling with her wristwatch, depending on the condition. We expected toddlers to imitate the sub-efficient means more faithfully in the wristwatch condition than in the smartphone condition. But there was no significant effect of condition on imitation of neither means nor goal. Thus, our hypothesis was not borne out by the results. In Experiment 2, using a within-subject design, we first assessed toddlers' ($N = 24$) performance in a no-demonstration baseline and then again after a no-disruption ostensive demonstration. In all three conditions with ostensive demonstration (Experiment 1: smartphone, wristwatch; Experiment 2: no-disruption), toddlers produced the demonstrated sub-efficient means significantly above the baseline level. In the no-disruption condition, goals were also imitated significantly above the baseline level. We conclude that the Sock Ball Task is a valid research tool for studying toddler imitation of novel means actions with objects. We end by discussing suggestions for improving the task in future studies.

## Introduction

When communicating with one another, adults rely on various sources to infer the speaker's intention to communicate [1, 2]. The richness of these sources and inferences is well captured

17605/OSF.IO/SJF4D; doi.org/10.17605/OSF.IO/E3W4Q).

**Funding:** The author(s) received no specific funding for this work.

**Competing interests:** The authors have declared that no competing interests exist.

in cases where the process eventually leads the recipient to abandon the initial stipulation about the speaker's communicative intention [3]. Imagine hearing a passer-by say "Hi. How are you?" as they approach you waiting at the bus stop. Even though they use a familiar phrase that typically opens a conversation, you may infer that they do not intend to communicate with you. What could support this conclusion? Perhaps while you turn to them with a confused smile, the rhythm and trajectory of their gait do not change at all. They rather avoid making eye contact with you than establish it. They burst out laughing and exclaim, "Yes, I am close. I am passing by the bus stop right now. Trzymaj się!" which–despite being true–is not what you would expect to hear next if they indeed had been addressing you: You do not see any reason for the laughter, the provided information has low relevance to you, and part of the message is in a language you do not speak. You eventually observe that they are wearing an earpiece, which confirms your suspicion that they are conversing with someone on the phone. You can even infer with some certainty that they are heading to a meeting and inform the interlocutor that they will be there soon. In this scenario, the inference is informed by the speaker's several verbal and non-verbal signals in relation to expectations evoked by (what could have been initially interpreted as) the opening ostensive addressing. It is also informed by context and semantic knowledge of earpieces and their use. It is an example of a complex inference of the kind adults routinely engage in as participants in ostensive-inferential communication [1]. But it is still unknown at what point in development children start engaging in this type of rich inferences about communicative intentions. In the current study, we sought to answer this question by investigating 18 – month-olds' imitation, when they are addressed by a model who uses a smartphone.

In the next three sections we will review the existing literature on the early sensitivity to communicative intentions in infancy, on smartphone use in face-to-face interactions in general and on its use in face-to-face adult-child interactions specifically. We will argue that the ubiquitous presence of smartphone use in face-to-face interactions creates a previously unexplored opportunity to further our understanding of toddlers' early inferences about communicative intentions.

## Early sensitivity to communicative intentions

There is a growing interest in the early developmental roots of pragmatic inferences [4–6]. Research on very young children's abilities to attribute communicative intentions has focused on their sensitivity to specific behaviors that act as *ostensive signals* for adults and elicit responses consistent with this function in infants. Newborns orient preferentially to eye contact [7]. Infants in the first months of life show a preference for contingent responsivity [8] and for sources of infant-directed speech [9, 10]–a prosodic pattern that typically signals communicative intention directed specifically at babies [11]. By 4.5 months infants preferentially orient to sound patterns of their own name [12]. By 5 months hearing their own name may have similar effects on neuronal activation and object processing as detecting eye-contact [13, 14].

Ostensive signals may facilitate interpreting other behaviors as communicative referential signals. For instance, infants around 6 months of age followed shifts of head and gaze with their own gaze, when these were preceded by infant-directed speech, but not when they were preceded by adult-directed speech [15, 16]. However, the results were mixed when the role of eye contact was assessed [16–18]. Eight-, 10- and 12-month-olds follow with their own gaze the orientation changes of a completely novel agent that first reacted contingently to the child's behavior [19–21]. By 12 months of age, pointing may be a referential gesture for infants, if preceded by communicative speech [22]. By 14 months, toddlers may rely on the previous shared experience with the speaker to interpret her ambiguous referential pointing, suggesting early pragmatic inference in an ostensive context [23].

We know much less about the early ability to infer communicative intention based on sources other than the early available set of ostensive signals. This seems to take time to develop in the first two years of life. According to one study, young 2-year-olds can interpret an intentional action of lifting a bucket by pulling a rope, as a communicative act even in the absence of typical ostensive and referential signals and language [24]. But how and when, during the first two years of life, children enrich their repertoire of pragmatic inferences supporting attribution of communicative intentions is largely unknown.

In the present study, we sought novel type of evidence for toddlers engaging in rich inferences about communicative intentions. Specifically, we hypothesized that by 18 months, toddlers might be able to recognize one commonly observed category of adult behavior as incongruous with ostension, namely adults' smartphone use in face-to-face interactions. Consequently, toddlers may rely on this interpretation when inferring the communicative intentions of adults.

## Smartphone use in face-to-face social interactions

Using smartphones during face-to-face interactions has become commonplace among the general urban population [25–28]. Despite its prevalence, such smartphone use is often perceived as having a negative impact on the quality of in-person interactions [26, 29–39]. Moreover, smartphone use in social interactions is often represented as socially unacceptable [40, 41] and as an impolite and annoying behavior [31, 35, 42–44]. These negative interpretations of mobile phone use are well captured by the term "phubbing", a portmanteau coined by a marketing campaign for the Macquarie Dictionary [45], reflecting how others' engagements with *phones* in face-to-face interactions may easily be interpreted as *snubbing* [28, 30].

Much of the literature examining the consequences of smartphone use in face-to-face interactions focuses on short- [26, 30, 31, 46, 47] and long-term [38, 48, 49] negative impacts on interpersonal relations and highlights the role of the negative feelings of being neglected and ostracized experienced by the partner, who does not use the phone [30, 40, 46, 47, 50–53]. Importantly for the topic of the current paper, several findings and theoretical themes in this literature are consistent also with the notion that adults detect how smartphone use during face-to-face interaction is at odds with what they expect from a communicative partner. It is often judged as going against various norms of social interactions [31, 35, 43, 54–58]. The person using the phone may be considered less responsive [46, 49, 59], inattentive to the partner [31, 35, 49], and not doing the due job of maintaining the common focus [49, 58, 60, 61]. Lack of access to the content that the phone-user focuses on is thought to affect the partner's response [62] and making the content of the smartphone a shared focus between the partners is considered a good strategy to remedy negative emotional impacts [58, 63]. Withholding eye-contact and contingent acknowledgements and responses is thought to be a typical part of phone use during in-person interactions [30, 53, 61, 64–66] and it is thought responsible for reflexive activation of feelings of being excluded and ostracized [30, 40, 53]. Notably, both eye-contact and contingent turn-taking are key behavioral signals regulating attribution of ostension and expectations related to them are indeed bound to be frustrated when the communication partner engages with a smartphone. For example, while eye contact is an important marker of shared attention throughout conversation [e.g., 67], the lack of or inconsistent eye contact caused by smartphone use signals inattentiveness [e.g., 35].

To our knowledge, the impact of the communicative partner's smartphone use specifically on the attribution of communicative intention and on related expectations has not been studied directly in adults. However, some factors thought to drive the emotional and interpersonal effects that have been studied are also likely to impact attributions of

communicative intention and to frustrate expectations about the communicative partner's behaviors and contributions [68].

## Smartphone use in face-to-face adult-child interactions

Just as smartphone use became commonplace in face-to-face interactions between adults, it became widespread in face-to-face interactions between adults and children. There is a growing body of literature exploring various aspects related to children's involvement in social situations where others use smartphones. In surveys, parents often report that their use of smartphones disrupts their face-to-face interactions with their children [69]. Observational studies have demonstrated that toddlers are frequently subjected to parents' smartphone use in environments such as restaurants, playgrounds, and waiting areas [70–76]. Moreover, the observational data show that in social situations with smartphones, children sometimes leave the parent-child interaction, misbehave, or express frustration or disappointment [71, 74]. Furthermore, McDaniel and Radesky [77] found an association between child behavior problems and parents' smartphone distractions in a survey using parental reports. Overall, smartphone use is assumed to impact parent-child communication negatively [71, 74, 77].

Some experimental studies hypothesized that adult phone use may impact learning negatively in young children. However, the results were mixed [78, 79]. For example, Konrad and colleagues [79] investigated the effect of parents' texting interruptions on toddlers' imitation learning and found no evidence for texting decreasing imitation rates. In contrast, Reed, Hirsh-Pasek, and Golinkoff [78] found that mothers' phone call disruption negatively affected toddlers' word learning, i.e., the toddlers showed no evidence of learning the novel words when the teaching session was interrupted by a phone call. However, as discussed by Reed, Hirsh-Pasek, and Golinkoff [78], it is unclear whether the underlying causes were specific to the phone disruption or whether they had to do with more general factors such as rate of eye contact, the mother's affect and body orientation, or the content of her communication. In this study, the lack of a control condition including a non-phone-related disruption precludes drawing firm conclusions about the specific effects of phone disruptions on learning in adult-child interactions.

Consistent with adults' negative responses to smartphone use during in-person interactions, several studies claimed that children tend to display negative emotions when they observe adults using smartphones [64–66, 80]. Modified still-face studies have shown that infants respond with increased negative affect when the parents pretend to use a smartphone [64–66]. However, because these studies did not use a matched control disruption without a smartphone, it remains unclear whether infants' negative responses were elicited by the appearance of smartphone use or if these were typical responses elicited in the still-face paradigm [for a review, see 81, 82]. Nevertheless, there is some evidence showing that young children react differently to phone disruptions than to other types of disruptions. For example, Rozenblatt-Perkal, Davidovitch, and Gueron-Sela [80] found a higher increase in infants' heart rates and negative affect (both of which can be interpreted as stress indicators) when a mother-child interaction was disrupted by maternal smartphone use compared to when the interaction was disturbed by the mother talking to someone present in person. It should be also noted that while several studies assume that still-face is a good model for adult behavior during smartphone use, it is not clear how well it approximates everyday smartphone use. Facial expressions during smartphone use do vary in frequency, valence, and intensity. For instance, a funny text message, or social-media content can make the reader frown, smile, or laugh [e.g., 83]. Furthermore, in real-life situations the degree of smartphone absorption varies depending on type of smartphone usage, e.g., quickly checking the time vs. reading a longer

text on the smartphone [e.g., 74]. Finally, by assuming still-face as a model of parental smartphone use, researchers focus primarily on the emotional impact on the infant, potentially foregoing the chance to explore the impact on the child's representation of the adult's behavior.

Potential impact of parental smartphone use on toddler emotional development and the interplay with factors such as parental stress and support of the child's autonomy on one hand, and children's needs, temperament and emotional competencies on the other remain an important topic for future research. However, it lies beyond the scope of the current paper. This broad topic and specific literature [84–88] were brought to our attention by an anonymous reviewer. We come back to it briefly in the discussion.

To summarize this short review, the current literature on adult smartphone use in adult-child interactions and its impacts is dominated by observational and parental-survey studies [69–77]. The conclusions from the few experimental studies are often severely limited due to the lack of necessary controls [78]. One commonly assumed model of parental behavior during smartphone use derived from the still-face paradigm, is likely not capturing the complexity of phenomena that infants are exposed to in real life [64–66]. Our approach in the current study was to go beyond these limitations by investigating the potential impact of adult smartphone-use on infant attribution of communicative intention, in an experimental design with carefully matched control. Furthermore, the study was driven by a theoretical proposal that went beyond the current literature reviewed above. We will present it in the next section.

## From exposure to adult smartphone use to early inferences about communicative intention: The hypothesis

When adults use their smartphones while interacting with young children, they are likely to behave in ways that are at odds with children's expectations about ostensive communication. We postulate that there are at least three ways in which this can happen.

First, studies have shown that smartphone use while being with children harms parents' contingency, sensitivity, and responsiveness to children's communicative bids [70, 71, 75, 89, 90]. Given young children's sensitivity to disrupted eye contact [91] and contingency [92, 93] in face-to-face interactions, inconsistent delivery of these ostensive signals during smartphone use is likely to affect their attributions of communicative intention.

Second, children are not able to identify the actual referent of many of the smartphone user's facial expressions and vocalizations. When these are taken as communicative behaviors produced for the child, this could lead to frustrated referential expectation [81, 94–96, for a related argument see 97]. Notably, in adult interactions engaging, with one's phone during a conversation leads to sharing the screen with the conversation partner only in a minority of cases [25, 63], and is presumably even less likely in adult-child interactions.

Third, if expectations related to relevance and common ground play a role at this age, these too may be frequently frustrated in interactions with a smartphone-using adult, whose communicative behaviors may often be delayed [70], have low or unclear relevance and provide poorly matched responses to the child's questions and requests. For example, Kelly and Ocular [89] found that smartphone-using parents more often reported being off-topic in conversations with their children at an aquarium than parents who did not use their smartphones. To sum up, we consider three ways in which adult phone use during in-person interactions with infants can go against infants' expectations in ostensive communication: (i) by crippling consistent delivery of ostensive signals and appropriate behavioral patterns, (ii) by failing to support fulfilment of referential expectation and (iii) potentially by frustrating burgeoning expectations of relevance and common ground.

Maintaining attribution of communicative intention to a smartphone-using adult may be further affected by how deeply absorbed they are with their device. It may be clearer–to children and adults alike–that someone deeply absorbed in reading something on their smartphone is not trying to communicate with them. On the other hand, when the use of the phone is interspersed with communicative contributions (or behaviors with a semblance of communicative contributions), it may become more challenging to determine whether one is the recipient of the ostensive communication. Some authors consider smartphone use in face-to-face interactions as "digital crosstalk" [65–67] akin to crosstalk, i.e., "a conversation or conversation-like activity maintained by persons who differentially share other interaction capacities" [Goffman 1963, as cited in 61]. Although this phenomenon is described for adults' face-to-face interactions with smartphone use, such crosstalk behaviors might as well be relevant to adults' interactions with children.

To sum up, (i) young children are frequently exposed to adult smartphone use in social situations [70–76]. (ii) Adults typically interpret such behaviors as counterproductive to high quality interactions and communication [26, 29–37]. (iii) Recent findings suggest negative emotional responses in young children to adult smartphone use [64–66, 80]. (iv) Our analyses suggested that adult smartphone use during in-person interactions may systematically frustrate infant expectations related to ostensive communication. Based on these premises, we hypothesized that children may from early on acquire a representation of smartphone use as incongruous with communicative intention. Furthermore, we expected that representing smartphone use this way may diminish children's certainty when attributing communicative intention to an adult who addresses them (i.e., signals intention to communicate) yet also uses a smartphone (i.e., engages in behavior incongruous with communicative intention). We chose to test this general hypothesis in toddlers around 18 months. By this age toddlers are known to consider a wider range of behaviors as communicative [98], to engage in early pragmatic inferences [23], and they may have ample experience with the disruptive effects of smartphone use in face-to-face communication.

In Experiment 1, we tested our general hypothesis, by relying on the assumption that ostensive communication facilitates imitation of novel means in toddlers [99–102]. We expected toddlers to imitate *less* faithfully, if the model, who addressed them ostensively, disrupted the demonstration by engaging in smartphone use, than if she engaged in a matched control behavior (fiddling with a wristwatch). In Experiment 2, we gathered data from two additional conditions: no-demonstration baseline and no-disruption condition, allowing us to assess the validity of our imitation paradigm further.

## Experiment 1

### Representation of smartphone use as incongruous with ostension

In Experiment 1, our main aim was to examine whether toddlers represent smartphone use as a behavior incongruous with ostension. If so, we expected them to be less certain (when compared to matched control participants) about attributing communicative intention to someone who addresses them ostensively but then uses a smartphone. To investigate this, we developed the Sock Ball Task—an online task assessing imitation of sub-efficient means and goal-outcome.

Sensitivity to ostensive signals is well established within imitation research [99–102]. It has been argued that in the imitation context, toddlers' sensitivity to the model's ostension allows them to infer that the model addresses them and through demonstrated actions conveys content that is relevant for them to learn [e.g., 103]. This relevance-guided interpretation of ostensively presented demonstrations, in which a model performs a goal-directed action, is thought

to support faithful imitation of demonstrated means actions, even if they are sub-efficient, i.e., not the most effective way of achieving the demonstrated goal [104, 105]. Consistent with this hypothesis, Király and colleagues [100] found that toddlers imitated sub-efficient means actions at a high rate (and significantly more often than when the context rendered the very same actions efficient) after having received them ostensively demonstrated, but did not show this pattern after having witnessed the actions performed without ostension [for a broader context of this study and different interpretations see also: 106, 107].

In line with these findings and theoretical perspective, we predicted that toddlers would imitate sub-efficient means actions less faithfully if an ostensive demonstration was disrupted by smartphone use compared to if it was disrupted by a matched control behavior (fiddling with a wristwatch). To investigate this, we assessed 17- to-19-month-old toddlers' immediate imitation of a goal-outcome and sub-efficient means actions. The toddlers watched a video demonstration of a model who ostensively addressed them and later non-ostensively performed the target actions with novel objects. Crucially, after the ostensive greeting but before the modeling of actions, the model either used a smartphone (smartphone condition) or fiddled with a wristwatch (wristwatch condition). We assumed that if the ostensive addressing was followed by behavior that toddlers represent as incongruent with ostensive communication, this would lower their certainty that the subsequent goal-directed action is part of the ostensive demonstration. Thus, if toddlers represent smartphone use (but not fiddling with a wristwatch) as incongruous with ostension and rely on this representation to infer others' communicative intentions, we expected toddlers in the smartphone condition to imitate the demonstrated means actions less faithfully on average than the toddlers in the wristwatch condition.

Data collection for this study was bound to happen during the unpredictable times of Covid-19 lockdowns. To test our hypothesis, we had to design an imitation task that could be implemented entirely online with both the experimenter and all the families participating from their homes. It was critical that matched sets of props could be created by parents only from the materials available at home. In this case the materials were: paper and socks. This is how the online Sock Ball Task was created.

## Materials and methods

### Preregistration

Experiment 1 was preregistered on the Open Science Framework on April 06, 2021 (osf.io/jy8av).

### Ethics statement

The study was ethically approved by the internal research ethics committee at the Department of Psychology, UiT The Arctic University of Norway (ref: 2017/1912) and by the Norwegian Centre for Research Data (NSD, ref: 973260). We obtained verbal informed consent from the participants' parents in an information meeting on Zoom.

### Participants

The participants were healthy toddlers between 17 months and 0 days and 19 months and 0 days. They were born without any complications after $\geq$ 37 weeks of gestation and with a minimum birth weight of 2500 g. All participating toddlers and parents understood Norwegian. We recruited the participants from all over Norway by advertising through kindergartens, health stations, libraries, social media channels and media coverage in magazines, newspapers, and in radio interviews. The participating families received a small gift and a certificate.

The required estimated sample size was $N = 48$ based on sample sizes from previous research [78, 79, 102, 108–111]. The final sample consisted of 48 toddlers ($M_{Age}$ = 546.98 days, $SD$ = 18.58, range = 519 to 578 days) who were randomly assigned to one of the two conditions: the smartphone condition ($n$ = 24, 12 girls, $M_{Age}$ = 548.25 days, $SD$ = 19.92, range = 519 to 578 days) and the wristwatch condition ($n$ = 24, 11 girls, $M_{Age}$ = 545.71, $SD$ = 17.48, range = 520 to 578 days). Sixteen additional toddlers participated but were excluded from the analyses due to failing to complete the test phase: no manipulation of test objects ($n$ = 8), major procedural disruptions such as extensive crying ($n$ = 3) or parental interference ($n$ = 5). None of the toddlers were excluded based on lack of attention to any of the important parts of the demonstration video in the demonstration phase.

## Design and procedure

The data collection started during the COVID-19 lockdown and lasted between 7th of April and 9th of November 2021. The families participated in the study from their homes via Zoom using a computer with a web camera and microphone. The experimenter conducted the experiment from her apartment.

An information meeting with the parent alone on Zoom was conducted to explain the study's purpose and procedure, to interview the parent about the inclusion criteria of the study and to obtain verbal informed consent. The information meeting was scheduled either for the day of the appointment or a day close to it.

Data collection was scheduled for the time of day when the parent reported that their child was most alert. The toddlers were seated on the parent's lap at a table, facing the computer placed outside their reach. The parent adjusted the camera angle to make the toddler and the table visible in the video. The parents hid their video-feed window to prevent toddlers from being distracted by their own images. The data collection session involved watching a demonstration video in a demonstration phase and meeting the experimenter live on Zoom for a warm-up phase and a test phase. The toddlers' behavior during all the phases was video recorded for later coding off-line.

**Demonstration phase.** Depending on the condition that each toddler was assigned to, in the demonstration phase they watched a video either with a smartphone disruption or with a wristwatch disruption (described in the section Demonstration videos and props). The parents shared their screens before playing the demonstration video, to allow for relating the video recording of the child's behavior to the content of the video stimuli. The live feed from the experimenter was not visible to the child and muted during the demonstration phase. The parent was instructed not to interact with and not to direct the toddler's attention to the demonstration video.

**Warm-up phase.** Immediately after the demonstration phase, the parent and the toddler met the experimenter, who had been the model in the demonstration video, live on Zoom. In her video feed the experimenter was sitting in the same setting, in the same position and wearing the same clothes as in the demonstration video. The experimenter and the parent talked briefly (approximately 30 seconds). During this interaction, the experimenter did not talk directly to the toddler but checked if the toddler was comfortable, as indicated by a lack of negative behavioral signs, such as crying or back arching. If the toddler showed any negative behavioral signs, the experimenter extended the warm-up by initiating a short (approximately 30 seconds) exchange with the toddler, during which the parent introduced the toddler to the experimenter, and the experimenter greeted the toddler by smiling and waving. This was done for 12 children in the final sample (i.e., 25%; 6 in each condition). The experimenter told the parent when the warm-up phase ended, and the test phase was to begin.

**Test phase.**  The parent put the test objects on the table when instructed by the experimenter. The experimenter pointed out that the toddler had the same test objects as her and said, "Look! We have the same things. [Toddler's name], it's your turn!". The test phase during which toddlers' imitation was scored offline lasted 1 minute counting either from the toddler's first touch of one of the test objects or from 30 seconds since all the test objects had been placed on the table, whichever came first. If the toddler did not touch the test objects during the first 15 seconds since placing the objects on the table, the experimenter repeated "[Toddler's name], it's your turn!". The experimenter repeated this instruction every 15 seconds until the toddler touched any of the test objects or until 1.5 minutes had passed since the objects were placed on the table. If the toddler showed any of the objects to the experimenter or the parent, the experimenter or the parent responded by saying, "Oh, how nice!". Fig 1 shows the procedure of Experiment 1.

## Demonstration videos and props

Each demonstration video started with music and text on the screen instructing the parent to be silent while the video was playing, followed by a colorful animation to attract the toddler's attention. Next, each video showed the same female model sitting behind the table. A smartphone was lying on the left corner of the table, and the model wore a wristwatch on her right wrist. The smartphone in the demonstration videos was a black iPhone 4S ($4.54 \times 2.31 \times 0.37$ inches), and the wristwatch was a black analog quadrangular watch sold by Clas Ohlson ($1.5 \times 1.5 \times 0.35$ inches). The same phone and wristwatch were visible throughout the demonstration videos and the warm-up and test phases when toddlers met the model live on Zoom.

The model looked directly into the camera and greeted the toddler by smiling, waving, and saying "Hi." This initial greeting was the only time when there were ostensive signals in either of the videos. It was immediately followed by a disruption, where the model either used the smartphone or fiddled with the wristwatch. In the smartphone condition, the model texted and swiped. In the wristwatch video, the model adjusted the length of the band around the wrist. The model's facial expressions during the disruption were kept neutral in both videos, with one little smirk in the middle of the disruption.

Immediately after the disruption, the model brought three objects from under the table, which she laid on the table-top: two A5 sheets of wrinkled white paper and a soft green ball made of a pair of socks. Then, she took the paper on her right side and covered the sock ball with it. Next, she grasped the sock ball through the paper. She lifted the paper-wrapped sock ball and moved it through the air to the left until it was above the other paper. Finally, the model dropped the sock ball out of the wrapping and kept the paper in her hand. The outcome of this action-sequence was the sock ball lying on the paper. Next the model took the test objects away from the table, put a new set of two papers and a sock ball on the table, and performed the same demonstration with them. At the end she again removed the three props from the table. Each video proceeded to a series of attention-getters presented in the four corners and in the middle of the screen to gather recording of the child focusing on these locations. Finally, a text was shown informing the parent that the video is finished and asking them to close the video link, stop screen-sharing and to put the Zoom session on their screen. Fig 2 displays still-frames from the demonstration videos. All the stimuli are available at OSF (doi.org/10.17605/OSF.IO/E3W4Q).

Parents were instructed (through a pre-prepared instructional video shared with them before the information meeting on Zoom) to prepare two sets of props. Each set consisted of two wrinkled (first squashed and then flattened out) A5-sized white paper sheets and a colored soft ball made of rolled socks. One was a spare set to replace damaged or out-of-reach objects

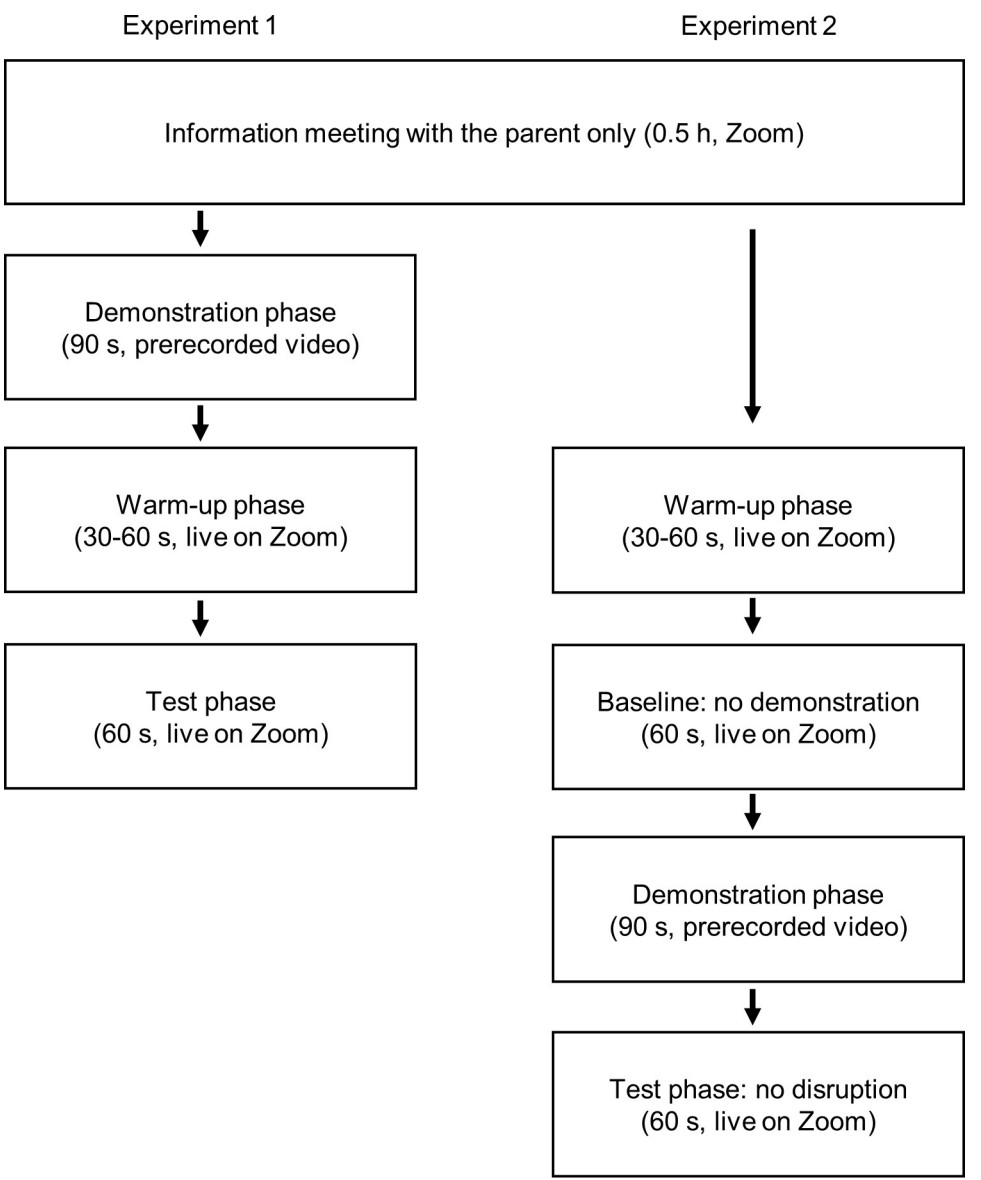

**Fig 1. Procedures of Experiments 1 and 2.** Note that the information meeting with the parent (uppermost row) always took place sometime before the data collection (the remaining rows).

during the test phase. The experimenter quality-checked all the test objects during an online information meeting before the data collection session.

## Coding

A naïve coder coded toddlers' looking to the demonstration videos to check for the exclusion criteria, i.e., that the toddlers saw at least 2 cumulative seconds of the ostensive greeting and the disruption, and at least one continuous demonstration of the action sequence.

We defined 4 action steps (cover, grasp, move, drop) and the main goal-outcome (ball on paper) to be coded from video recordings of the test phase. We operationalized each of the four action steps by specifying a sub-goal for each of them together with minimal specific criteria for the child's behavior, which would nevertheless help distinguish the target action steps

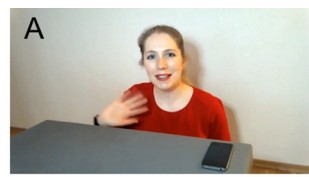

Ostensive greeting

Smartphone disruption

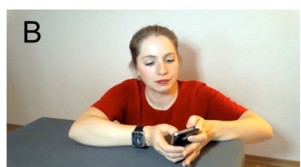

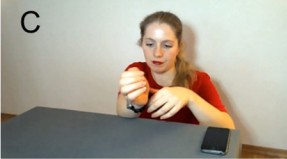

Wristwatch disruption

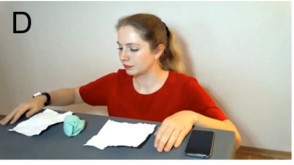

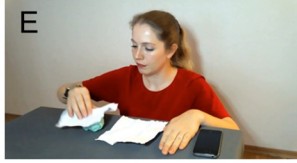

Cover

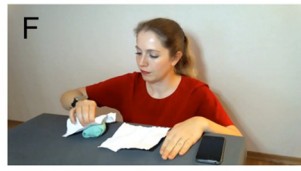

Grasp

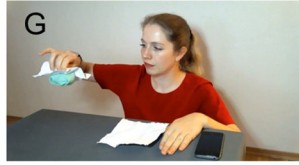

Move

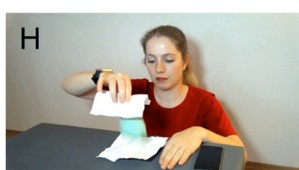

Drop

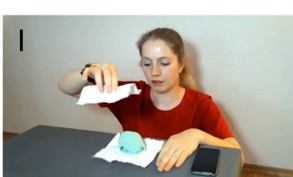

Goal-outcome: Ball on paper

**Fig 2. Stimuli in the demonstration phase.** Still frames from the stimuli used in the demonstration phase. In Experiment 1, the stimuli for the two conditions (smartphone vs. wristwatch) were closely matched but differed in whether they showed smartphone disruption (B) or wristwatch disruption (C). After producing the goal-outcome (I), the model removed the three props from the table, put another identical set on and performed the sequence shown in still frames D-I once more. In Experiment 2, stimuli used in the demonstration phase of the no-disruption condition were closely matched to those in Experiment 1 but consisted only of steps A and D-I, with the sequence D-I also performed twice.

**Table 1. Coding scheme for the four action steps and the final goal-outcome with specified sub-goals, operational definitions, and additional coding instructions.**

| Action step /final goal-outcome | Sub-goal, operational definition, and additional coding instructions |
|---|---|
| Cover | **The paper is on the sock ball.** *The toddler places the paper on top of the sock ball so that it covers all of the top.* It does not matter where the sock ball is, e.g., on the table or in hand. Code "yes" if the ball was already in contact with the paper before it got into the position where the paper was on top of the ball. It is OK if "cover" results from grasping the ball. |
| Grasp | **The sock ball is being held through the paper.** *The toddler grasps and holds the sock ball so that at least part of the paper is between the hand and the sock ball.* |
| Move | **The location of the sock ball is changed by using paper.** *The toddler changes the location of the sock ball by making contact between the ball and the paper.* This is regardless of the length and direction of the movement, as well as the relative positions of the paper and ball (i.e., changing the location of the sock ball with the paper on top, underneath, or from the side). The child must act on the paper, not just on the ball. Although there is no explicit lower limit on the length of movement, micro-movements should not be coded as "yes." |
| Drop | **The sock ball is being dropped down.** *The toddler drops the sock ball by letting go of the ball before the ball touches a surface.* Throwing forward/away also counts as dropping (i.e., the fall doesn't have to start downwards), but throwing the ball up doesn't. |
| Ball on paper | **The sock ball is on one of the papers.** *The toddler makes contact between the sock ball and the paper, with the sock ball on top of the paper.* The exact means by which this goal is achieved (e.g., by putting, dropping, rolling, or banging it on the paper) does not matter. It doesn't matter whether the toddler lets go of the ball (e.g., opens their hand) once the sock ball is on the paper. It does not matter where the paper is when the ball is on it. Code as "no" if the outcome is a byproduct of handling the ball and paper together but do code as "yes" if it is an intended outcome. Code as "yes" if not all the ball is on the paper. |

*Note*. Sub-goals are in **bold**. Operational definitions are in *italics*. Additional coding instructions are in plain text. The same coding scheme was used in both experiments.

from superficially similar unintended events (Table 1). We formulated the operationalizations based on our analyses of the experimenter's action in the demonstration video (Fig 2) as well as the actual actions produced by toddlers participating in the pilot.

In formulating the operationalizations, we were guided by two general assumptions: First, we assumed that any emulation of the predefined sub-goals of the action steps (and the main goal-outcome) should count if it was successful (i.e., the sub-goal was produced), and if it satisfied the specific minimum criteria. Second, we assumed that production of any of the sub-goals (or the main goal-outcome) should count regardless of how long the state lasted (e.g., how long the paper stayed on the sock ball, or the sock ball stayed on the paper). Except for the *move* action step, in which case micromovements were to be ignored.

Children's behaviors during the test phase were coded offline from videos recorded at 25 frames per second viewed frame by frame using media player software (mpv). Each instance when the child performed a behavior falling into one of the coded categories was coded in the order of occurrence. There were two coders blinded to the condition that the toddlers were assigned to: the second and the last author. They first coded randomly chosen 21 videos (i.e., approximately 1/3 of the videos from all 65 toddlers tested) independently and reached excellent inter-rater reliability ($\kappa = .953$) on a total of 105 yes or no decisions about toddlers' target action performance (21 videos × (4 action steps + 1 goal state)). Next, they coded the remaining 2/3 of the videos together, resolving disagreements through discussion. Note that this procedure deviated from the preregistered coding procedure, where we planned for two independent coders blinded not only to the condition but also naïve to the

study's hypothesis. The reason for this deviation was that we could not ensure that the same naïve coders would be available to code the data in Experiment 2. Consequently, we opted for the two authors who did not perform the data collection to code the videos while ensuring that they remained blinded to the condition (before coding, the videos were pre-edited by the first author, who collected the data, to remove any indicators of the experimental condition).

We computed three dependent measures based on offline coding: faithful imitation score, binary means score, and binary goal attainment score.

The faithful imitation score ranged from 0 to 8. It was the sum of the number of types of action steps produced (range: 0–4), and the best-sequence score (range: 0–4). For calculating the best-sequence score, each action step produced during the test phase received a numerical code according to its place in the modeled action sequence (cover = 1, grasp = 2, move = 3, drop = 4, main goal-outcome = 5). The best-sequence score was equal to the number of steps in the longest-produced sequence of ascending codes, minus one. Note that this procedure for calculating the faithful imitation score differed from our preregistered definition. See S1 Appendix for more details and for the results obtained using the preregistered definition.

The binary means action score equaled 1 if the child produced any of the target action steps at least once during the test phase. Otherwise, it equaled 0. The binary goal attainment score equaled 1 if the child produced the final goal-outcome at least once during the test phase. Otherwise, it equaled 0.

## Results

SPSS (Statistical Package for Social Sciences 28.0) was used for all statistical analyses. For all statistical tests, p-values are reported for exact two-tailed tests. The distribution of the faithful imitation score was significantly different from normal, as indicated by a Kolmogorov-Smirnov test, $D(48) = 0.284$, $p < .001$. The skewness of the faithful imitation score was 1.22, indicating that the distribution was right-skewed. Therefore, non-parametric tests were used to analyze this variable.

There was no statistically significant difference in the faithful imitation score between the smartphone condition, $Mdn = .50$, range = 0–8, and the wristwatch condition, $Mdn = 1$, range = 0–7, Mann-Whitney $U(N_{smartphone.} = 24, N_{wristwatch.} = 24) = 238.50$, $z = -1.07$, $p = .291$, $r = -.15$ (see Fig 3). Since, as indicated in the methods section, the analysis reported here was based on a definition of faithful imitation score that differed from the definition preregistered for Experiment 1, please note that the overall pattern of statistical significance remains the same if the preregistered definition was to be used. For details, see S1 Appendix.

In total, 13 toddlers in the smartphone condition and 12 in the wristwatch condition produced the main goal-outcome, $p = 1.00$ (Fisher's exact test, $N = 48$) (see Fig 4). Twelve toddlers in the smartphone condition and 16 in the wristwatch condition produced at least one type of action step, $p = .380$ (Fisher's exact test, $N = 48$). The number of children producing means (i.e. action steps) over the final goal was not statistically different across the two conditions, as assessed by Generalized Estimating Equations with binary logistic model and an exchangeable covariance matrix structure on the binary means action score and the binary goal-attainment score, Wald Chi-Square test, $W(1) = 1.746$, $p = .186$.

## Discussion

The findings from Experiment 1 were not consistent with our main hypothesis. We found no evidence that toddlers who watched the demonstration disrupted by smartphone use, imitated

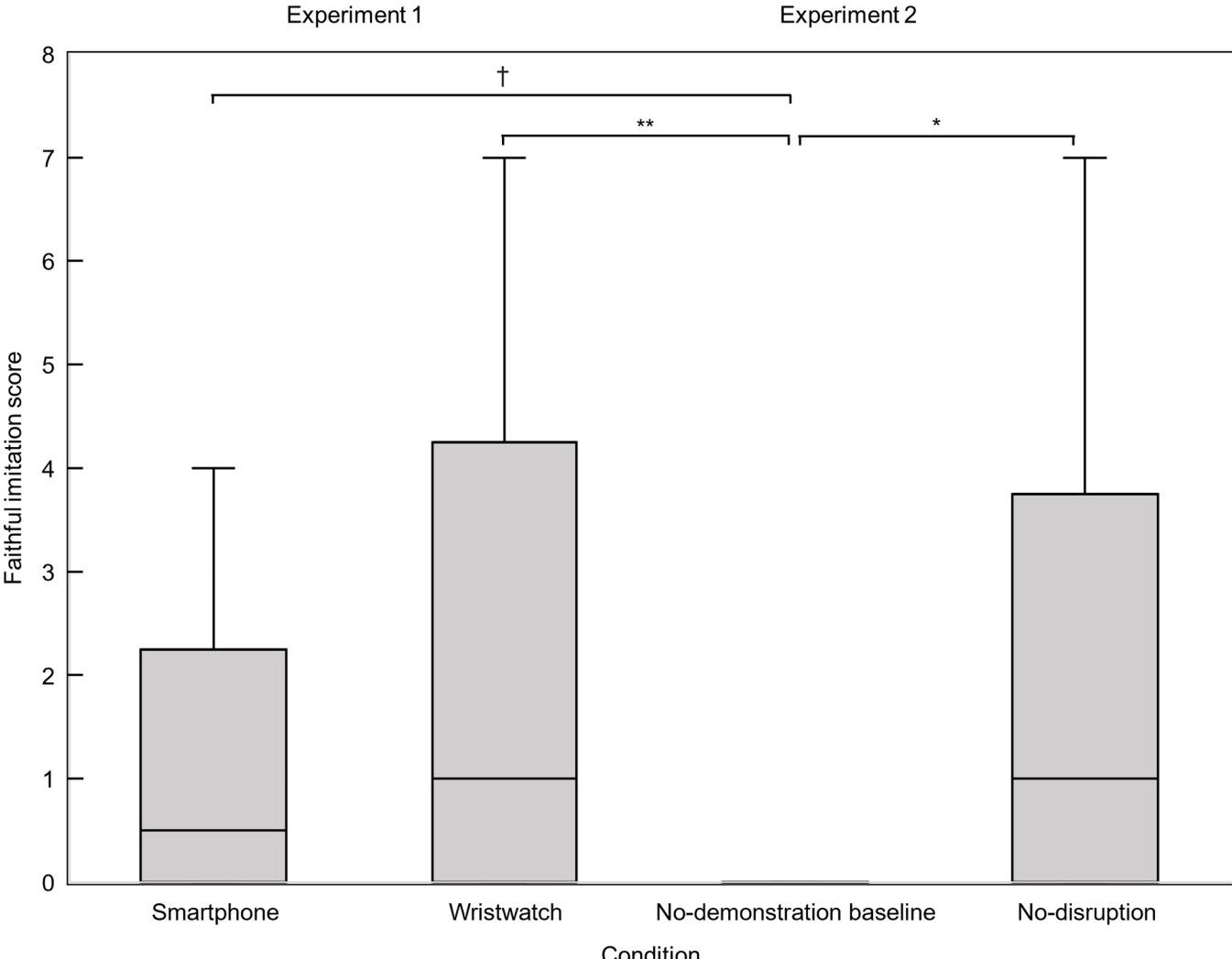

**Fig 3. Faithful imitation scores in Experiments 1 and 2.** Distribution of faithful imitation scores in the smartphone and wristwatch conditions of Experiment 1, and no-demonstration baseline and no-disruption conditions of Experiment 2. Each boxplot indicates condition median, midspread, and 1.5 interquartile range. Data points with values above 1.5 interquartile range not shown. * $p < .05$ and ** $p < .001$ by exact two-tailed Wilcoxon signed-rank test. † $p < .05$, by exact two-tailed test Mann-Whitney test.

the modelled means less faithfully than those who watched the demonstration disrupted by fiddling with a wristwatch. Thus, our hypothesis that toddlers represent smartphone use as incongruous with ostension and rely on this representation when inferring communicative intention, was not supported by the results of Experiment 1.

Null results are notoriously difficult to interpret. They may reflect lack of the hypothesized effect. But they may also stem from problems in, e. g., the operationalization of the hypothesis and in the measurement. We considered two such explanations of the results of Experiment 1.

First, the test of the hypothesis in Experiment 1 relied on the assumption that the faithful imitation score indeed measures imitation. One explanation for why there was no hypothesized difference between conditions in Experiment 1 is that contrary to our assumption, the procedure might have not measured imitation but rather spontaneous production of the target behaviors, possibly unrelated to the demonstration videos. The design of Experiment 1 alone could not provide data in support of the assumption that the faithful-imitation score measured imitation, because the baseline level of this key measure was not assessed.

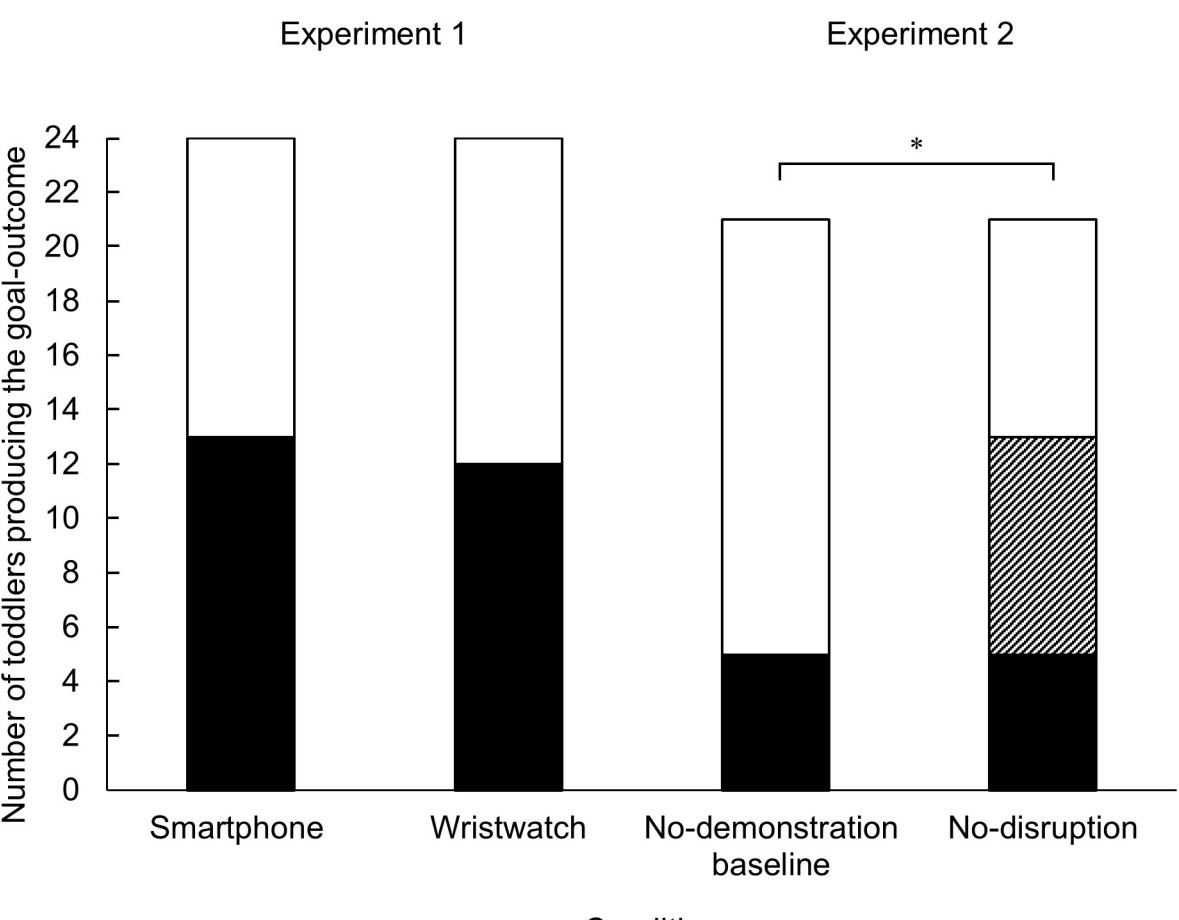

**Fig 4. Goal-outcome imitation in Experiments 1 and 2.** Number of children producing the final goal-outcome in the smartphone (*n* = 24) and wristwatch (*n* = 24) conditions of Experiment 1 (between subject design), and no-demonstration baseline (*n* = 21) and no-disruption baseline (*n* = 21) conditions of Experiment 2 (within-subject design). Note that in Experiment 2 only children who produced valid data in both conditions were included in this analysis. White bars indicate toddlers who did not produce the goal-outcome. Black bars indicate toddlers who produced the goal-outcome. Stripped bar indicates children who produced the goal-outcome only in no-disruption condition, but not in baseline. * *p* <. 05 by exact two-sided McNemar's test.

Second, if toddlers indeed imitated and our measure indeed captured their imitation, the effect of the type of disruption on imitation might have not been detected in statistical tests, because the procedure was ill-suited to capturing it. This could have been the case, for instance, if some elements of the procedure common to both conditions were affecting toddlers' imitation negatively in both conditions to start with. The disruption of the demonstration might have impacted imitation, both when it involved phone use and when it did not, for instance, because it led toddlers to process first the model's actions on an irrelevant object (the smartphone, the wristwatch), which later was not available to them to act on during the test phase. Knopf and colleagues [112] reported that when the presentation order of items differed between the demonstration and the test phase in a deferred imitation task, 10-to-11-month-old infants' imitation performance decreased. Task and age-group differences aside, one can speculate that if the toddlers in the current study encoded the smartphone and the wristwatch as first objects in the demonstrations, then their imitation performance might have been impacted, as a result, because the test phase did not start with these objects. Furthermore, imitation levels in Experiment 1 could have been dampened down because the participants in

both conditions had little interaction with the experimenter prior to the test phase. For example, Kim and colleagues [113] found that 18-month-old toddlers were more likely to imitate after interacting with the experimenter in a warm-up phase prior to the imitation test, compared to a warm-up phase where the experimenter did not interact with them.

Experiment 2 was conducted to address these two concerns.

## Experiment 2

### Assessing baseline performance and imitation after an undisrupted demonstration

The aim of Experiment 2 was to gather empirical evidence that could further inform the interpretation of the null result of Experiment 1. Specifically, we assessed toddlers' performance using the same Sock Ball online task as in Experiment 1 but in two new conditions: no-demonstration (baseline) and no-disruption (Fig 1). As these labels suggest, in the no-demonstration condition we assessed toddlers' spontaneous production of the key behaviors. In the no-disruption condition toddlers' performance was assessed after they watched a demonstration video. Like in Experient 1, the video showed the ostensive greeting followed by a non-ostensive demonstration of the modelled actions. But unlike in Experient 1, the model proceeded to the demonstration immediately after greeting the toddler (Fig 2). Importantly, each child participated in both conditions in a fixed order: no-demonstration baseline first, no-disruption condition second (Fig 1). Thus, the procedure of the baseline condition closely matched the procedure of the two conditions in Experiment 1, except for the lack of demonstration video. The procedure in the no-disruption condition also closely matched those in Experiment 1, except for the different video stimulus (Fig 2) and for the fact that by the time of the test phase toddlers had extensive opportunity to warm up to the experimenter during the preceding baseline and that they manipulated the target objects in the baseline condition.

The main preregistered hypothesis for Experiment 2 was the following. If our Sock Ball online task indeed measured imitation, we expected that the goal-outcome would be produced less frequently in the baseline than in each of the conditions of Experiment 1. Furthermore, we anticipated that the faithful-imitation score would be significantly lower in the baseline than in the wristwatch condition of Experiment 1, where we had expected toddlers to express unaffected imitation of the modeled means. Likewise, we also expected both production of the goal-outcome and of the modelled means to be significantly lower in the baseline than in the no-disruption condition. Furthermore, if factors such as the lack of extended warm-up and the presence of task-irrelevant object-use indeed dampened toddlers' imitation in Experiment 1, one would expect the goal-outcome production and the faithful-imitation of means to be significantly higher in the no-disruption condition than in the two conditions of Experiment 1. Because our preregistered "stopping rule" was tied to completing the full sample for the baseline, all the comparisons involving the no-disruption condition were preregistered as secondary analyses.

## Materials and methods

### Preregistration

Experiment 2 was preregistered on the Open Science Framework on August 17, 2022 (osf.io/jytrg).

### Ethics statement

The study was ethically approved by the internal research ethics committee at the Department of Psychology, UiT The Arctic University of Norway (ref: 2017/1912) and by the Norwegian

Centre for Research Data (NSD, ref: 973260). We obtained verbal informed consent from the participants' parents in an information meeting on Zoom.

## Participants

The participants were healthy 17- to 19-month-old toddlers, recruited in the same manner as in Experiment 1. Each participant was assigned to both the no-demonstration baseline condition and to the no-disruption condition. Data collection ended when the preregistered sample size (after exclusions and replacements) was reached in the baseline condition. Thus, the final sample in the baseline condition was $N = 24$ toddlers (14 = girls, $M_{Age} = 548.21$ days, $SD = 19.18$, range = 518 to 577 days). Three more participants were tested in the baseline condition but excluded from the analyses because of parental interference (1), incorrect test objects (1), and for not completing the baseline phase and no manipulation of the test objects (1). The final sample in the no-disruption condition was $N = 22$ toddlers (12 = girls, $M_{Age} = 547.91$ days, $SD = 18.72$, range = 518 to 577 days). Twenty-one of these toddlers had valid data for both the baseline and the no-disruption conditions. Five toddlers were tested in the no-disruption condition but excluded from the analyses due to technical error (1), parental interference (1), or not completing the test phase (no manipulation of test objects ($n = 1$) or major procedural disruptions such as extensive crying ($n = 2$). All participants met the inclusion criteria for attention in the demonstration phase of the non-disruption condition. The data collection was conducted between August and November 2022.

## Design and procedure

The data collection lasted between 23rd of August and 5th of November 2022. Each toddler participated first in the baseline condition and next in the no-disruption condition. The procedures for these two conditions differed from that of Experiment 1 in the following ways (Fig 1). In the baseline condition there was no demonstration phase. The session started with a warm-up phase. Similar to Experiment 1, for 4 out of 24 toddlers (16.7%) the warm-up phase was extended with a short toddler-experimenter interaction. Warm-up was followed immediately by a test phase. Next, in the no-disruption condition there was a demonstration phase, followed by a test phase.

## Demonstration videos and props

The props and the procedure for preparing them by the parents were the same as in Experiment 1. The experimenter (who was also the model in the demonstration video) was also the same as in Experiment 1. The only difference was that in the video shown in the demonstration phase of the no-disruption condition the model produced ostensive greeting and proceeded immediately to the non-ostensive demonstration (Fig 2).

## Coding

The coding procedure was identical to the one used in Experiment 1, and it had been preregistered for Experiment 2. The second author coded toddlers' looking to the demonstration videos to check for the exclusion criteria. The second and the third author, blinded to the condition, coded the participants' behavior during the test phase together and resolved differences through discussion.

## Results

SPSS (Statistical Package for Social Sciences 28.0) was used for all statistical analyses. For all statistical tests, p-values are reported for exact two-tailed tests. A Kolmogorov-Smirnov test was conducted to assess the normality assumption on the faithful imitation score, and it indicated that the distribution of the faithful imitation score was significantly different from normal, $D(94) = 0.297$, $p \leq .001$. The distribution was right skewed as indicated by a positive skewness value of 1.36. Therefore, non-parametric tests were used to analyze this variable.

The faithful imitation score was significantly lower in the baseline condition, $Mdn_{baseline} = 0$, range$_{baseline}$: 0–6, than in the wristwatch condition of Experiment 1, $Mdn_{wristwatch} = 1$, range$_{wristwatch}$: 0–7 (Mann-Whitney test, $U[N_{baseline} = 24, N_{wristwatch} = 24] = 151.50$, $z = -3.12$, $p = .001$, $r = -.45$). The faithful imitation score was also significantly lower in the baseline condition, $Mdn_{baseline} = 0$, range$_{baseline}$: 0–6, than in the smartphone condition of Experiment 1, $Mdn_{smartphone} = .50$, range$_{smartphone}$: 0–8 (Mann-Whitney test, $U[N_{baseline} = 24, N_{smartphone} = 24] = 200.50$, $z = -2.12$, $p = .035$, $r = -.43$). This post-hoc comparison was not preregistered because of lack of clear hypothesis. We report it here for completeness.

On the other hand, a Fisher's exact test indicated that the production of the goal-outcome was not significantly different between the baseline and the wristwatch condition ($p = .135$). Moreover, a Fisher's exact test indicated that the production of the goal-outcome was not significantly different between the baseline and the smartphone condition ($p = .075$). In total, 6 out of 24 toddlers produced the goal-outcome in the baseline condition, whereas 12 toddlers did in the wristwatch condition and 13 toddlers did in the smartphone condition.

Of the 24 toddlers in the final sample, $N = 21$ toddlers had valid data for both baseline and no-disruption conditions. The faithful imitation score was significantly higher in the no-disruption condition ($Mdn = 1$, range = 0–7) than in the baseline ($Mdn = 0$, range = 0–6), Wilcoxon signed-rank test, $z = -2.16$, $p = .029$, $r = -.47$ (See Fig 3). Furthermore, there was a statistically significant difference in production of the main goal-outcome between the baseline condition and the no-disruption condition (McNemar's test, $p = .008$). Of the 21 toddlers who provided valid data in both conditions only 5 (24%) produced the goal-outcome in the no-demonstration baseline condition (1 additional child also did but was excluded from the no-disruption condition) with none of them doing so in the baseline only. On the other hand, 13 (62%) of the 21 produced the goal-outcome in the no-disruption condition with 8 of them (i.e., 62% of the goal producers in this condition and 38% of the whole sample of 21) produced the goal only in the no-disruption condition but not in the baseline (See Fig 4).

There were no other statistically significant differences in the production of faithful imitation scores nor in goal production between the no-disruption condition and neither the smartphone nor the wristwatch conditions of Experiment 1. More specifically, the performance on the faithful imitation score was not significantly different between the no-disruption condition ($Mdn_{no-disruption} = 1$, range$_{no-disruption} = 0–7$) and the smartphone condition ($Mdn_{smartphone} = .50$, range$_{smartphone} = 0–8$), Mann-Whitney test, $U[N_{no-disruption} = 22, N_{smartphone} = 24] = 240.00$, $z = -.56$, $p = .583$, $r = -.08$), nor between the no-disruption condition and the wristwatch condition, Mann-Whitney test, $U[N_{no-disruption} = 22, N_{wristwatch} = 24] = 244.50$, $z = -.45$, $p = .663$, $r = -.07$). Fisher's exact tests indicated no statical difference in goal production between the no-disruption condition and the smartphone condition ($p = .774$), nor between the no-disruption condition and the wristwatch condition ($p = .568$).

## Discussion

The aim of Experiment 2 was to gather further evidence to inform the interpretation of the null results of Experiment 1. The results leave little doubt that the online Sock Ball Task indeed

captured—as assumed—toddlers' imitation of the modelled means. This is evident in the faithful imitation score being lower in the no-demonstration baseline condition than in the wristwatch condition (preregistered primary comparison), the no-disruption condition (preregistered secondary comparison) and in the smartphone condition (post-hoc comparison). The results concerning imitation of the main goal-outcome were less clear. Even though production of the goal-outcome was numerically lower in the baseline than in all the other three conditions, only the within-subjects comparison between the baseline and the no-disruption condition brought a statistically significant result.

The lack of statistically significant differences in imitation between the no-disruption condition of Experiment 2 and neither of the conditions in Experiment 1 speaks against the possibility that the imitation levels in Experiment 1 had been dampened down because of insufficient warm-up or because of exposure to task-irrelevant object-use during the disruptions.

## General discussion and conclusions

In two experiments, we assessed toddlers' imitation of novel means actions as well as goal-outcomes using a new Sock Ball Task conducted entirely online. Experiment 1 was designed to test the hypothesis that toddlers represent smartphone use as incongruous with ostensive communication and rely on this representation when inferring communicative intention. Consistent with this hypothesis and in accordance with previous literature, we expected toddlers to imitate the novel means less faithfully when ostensive demonstration was disrupted by smartphone use, than when it was disrupted by a matched control behavior. We did not find the expected difference. Comparisons to the baseline performance assessed in Experiment 2 confirmed that as a group, 17-to-19-month-olds imitated the novel means actions presented on the video. Moreover, their faithful imitation was not affected by factors which in principle could have dampened it down, such as the length of warm-up and the presence of task-irrelevant information. We conclude that: (i) the Sock Ball Task is a valid tool for assessing toddlers' imitation of novel means-actions. (ii) The current study found no empirical support for the initial hypothesis regarding toddlers' representation of smartphone use.

In what follows we consider several ways in which these results can be reconciled with the general pattern of findings suggesting that in adults smartphone use in social interactions goes against conventions and expectations of well-formed communicative interactions [29–37].

### The age group

Could the 1.5-year-olds participating in our study be too young an age-group for testing our hypothesis? We do not find this plausible. The effect of ostension on imitation of novel means has been reported even in younger children [100]. Furthermore, numerous studies suggest that 1.5-year-old toddlers likely have ample experiences with communication partners being distracted by smartphone use [70–76]. These experiences might have been even more frequent during the COVID-19 pandemic. Studies using retrospective parent reports suggest that many parents increased their smartphone use compared to pre-pandemic smartphone use [e.g., 114, 115].

Could the 1.5-year-olds then be too old of an age group for testing our hypothesis? This we find more plausible for two reasons. First, although various studies suggested that ostensive demonstration may facilitate imitation of sub-efficient means actions in toddlers between 14 and 18 months of age [99–102], there are also reports of age trends in imitation of the sub-sufficient means. For instance, situational constraints were found to affect imitation of ostensively presented means in 12- and 14-month-olds [100, 116]. But at 18 months, the effect was much

weaker and seemed to vane in older children, who tended to imitate more overall and to a similar extent both when the situational constraints rendered the action sub-efficient with respect to the goal, and when they did not [117]. Thus, currently, the existing literature does not give a clear picture about the interplay of communicative context and sub-efficiency of the modelled actions on the imitation around 1.5 years of age. In hindsight, a younger age group, e. g., 14-month-olds, might have been a better choice for testing the hypothesis of the current study.

Second, because smartphone use is omnipresent in everyday interactions, adults tend to habituate to face-to-face interactions disrupted by smartphone use [37, 47, 118, 119], and so could infants and toddlers. It should also be noted that adults' opinions about smartphone use related disruptions are not uniformly negative, but rather depend on context [120], type of usage [43, 120, 121], attitudes [47], and quantity of smartphone disruptions [25]. Moreover, age might play an important role in the perceived appropriateness of smartphone use in face-to-face interactions [121]. For example, Forgays and colleagues [121] found that younger adults tend to perceive smartphone texting in various social scenarios as more acceptable compared to middle-aged and older individuals.

Adults can learn to adjust their expectations about communicative face-to-face interactions based on experiences with individual smartphone users [28, 47, 122] and it seems likely that children may go through a similar process in the developmental perspective. Cross-cultural comparisons suggest that the processes that allow infants to "tune into" the culture-specific interactional patterns start very early on in ontogeny [123]. Furthermore, one study [65] found no association between mothers' self-reported smartphone habits and infants' behaviors in a modified still-face phase with a mobile device. As discussed in the paper, the lack of association could be explained by infants adjusting to their environment. Relatedly, children's past experiences with how parents respond to their behavior in emotionally demanding situations, such as parental scaffolding and autonomy support, might foster emotional self-regulation (e.g., ability to wait) when children encounter similar events in the future [85, 88, 124]. We can speculate that in environments where smartphone use is frequent in face-to-face adult-child communication, infants learn the smartphone-use-heavy interactional patterns present in those environments from early on. On this account, at 1.5 years toddlers in the smartphone condition of the present study might have been well attuned to treating ostensive addressing followed by smartphone use as a straightforward case of ongoing communication directed at them. Younger toddlers, or even infants, could be a more suitable age-group to test the hypothesis that smart-phone use is represented as incongruous with ostensive communication, as they are earlier in the process of learning the behavioral patterns of communication common in their environments and attribution of communicative intention may rely more on the developmentally prior sensitivity to ostensive signals and on the expectations that they elicit when detected [e.g., 125].

## The procedure

Toddlers in the no-disruption condition of Experiment 2 imitated to a comparable degree to those in the smartphone and the wristwatch conditions of Experiment 1. Thus, we found no evidence for the role of some procedural factors that in principle could have affected imitation levels *negatively*. But perhaps some aspects of our procedure could have affected performance *positively*, and consequently made it harder to detect the effect of interest?

Some recent studies suggest that for young children one function of imitation is facilitating social affiliation [126, 127]. For the toddlers in our study, imitating the modeled actions might have been a way of communicating and affiliating with the model on the computer screen. Crucially, this function was available in all conditions except for the no-demonstration

baseline. At this point we can only speculate what aspects of the procedure, common across conditions, might have played a role in facilitating it. We note here two possible factors. (i) Because of the prerecorded demonstration stimuli, the test-phase was the first opportunity to interact with the model. Before that, during the demonstration phase, many toddlers might have tried unsuccessfully other ways of communicating with the pre-recorded model or attracting her attention through vocalizing or waving. Anecdotal observations indicated that some of the toddlers greeted and waved back when watching the demonstration video of the model's ostensive greeting. (ii) As reported, most toddlers did not need any extra warm-up with experimenter-child interaction before the test phase. Consequently, for most participants the test phase was preceded by an interaction between mostly the experimenter and the parent, that excluded the child. This might have mattered given, e.g., that imitation fidelity in young children increases when exposed to ostracism [128–130]. Notably, smartphone disruptions in face-to-face interactions are associated with feelings of being ostracized in adults [32]. If they evoke similar emotional response in toddlers, then on this account they might have in fact motivated the toddlers in the smartphone condition to imitate for affiliative gains, thus counteracting the effect we had originally hypothesized. Future studies building on the current paradigm may take these hypothetical factors into consideration.

## Limitations, strengths, and reflections on online testing

We want to point out some further limitations of our study. One of the assumptions behind our hypothesis was that infant experiences with adult smartphone use in everyday life prior to participating in the study drove their representations of smartphone use. However, participants were recruited from the general population and there was no assessment of the actual exposure of participating toddlers to adult smartphone use. This was done for two reasons. First, because data had to be collected during the Covid-19 pandemic, the available measures of smartphone use would be parental self-reports [28, 39, 131–133] or data from phone-use tracking applications [134–136]. Self-reports are known to be unreliable and highly susceptible to social desirability factors [137], and studies suggest that adults' retrospective self-reports of their own smartphone use can be susceptible to both underestimation and overestimation [135, 136, 138–140]. By a similar token, we reasoned that asking parents to use an application that tracks their phone-use is likely to affect their typical smartphone habits. Second, by now many studies document high levels of smartphone use in everyday parent-child interactions [70–76]. Moreover, it has been suggested that adults often use their smartphones during in-person communication with those who are closest to them [27]. Therefore, we assumed that as a group, participating toddlers likely had sufficient experiences with this behavior. But the procedure did not allow us to verify this assumption.

Another limitation is related to how we operationalized smartphone use in the demonstration stimuli. We tried to mimic how smartphone use disruption may occur in a real-life social interaction. But neither the length of the smartphone disruption (10 seconds) nor the details of the action (swiping and texting) were derived from any real-life data on typical adult behaviors, nor independent interpretation of this behavior by adult viewers. Of course, in doing this we followed in the footsteps of many developmental researchers before us, who also relied on common sense judgement and intuition, when designing the stimuli for infants and toddlers. Still, we want to acknowledge this as one of the limitations of the current attempt.

A further point to note is that both our general hypothesis and its proposed test through means-imitation task were designed for typically developing children, and it is uncertain whether they may generalize to non-typically developing populations and contexts, for at least two reasons. First, imitation behaviors involving novel actions on objects can vary significantly

between typically and non-typically developing children [e.g., 141]. Second, children's prior exposure to adult smartphone use could be influenced by parental factors such as maternal depression [142]. Notably in the current study we did not collect any data about the parenting context. Lastly, all the participating toddlers and families were Norwegian speakers living in Norway, thus our sample is not representative of the broader population. Although smartphone ownership is increasing globally, with one internet source estimating that 85% of the global population owns a smartphone [143], the majority of people in developing countries do not have a smartphone [144]. In contrast, but similar to many other Western countries [143], in Norway, the majority (98%) of the population (between 9–79 years) is estimated to have access to a smartphone [145].

On the other hand, we consider it a valuable contribution of the current study that it introduces a new online imitation paradigm, the Sock Ball Task. We developed the task during the COVID-19 lockdown, which restricted data collection in the lab. This limitation forced us to design an imitation task for which reasonably matching sets of props could be produced by individual participating families with materials available to them at home: two wrinkled A5 sheets of paper and a pair of socks rolled into a ball. We found evidence for imitation of novel sub-efficient means actions in three separate groups of 17-to-19-month-olds. Evidence for the imitation of goal-outcome is less clear as it was found in only one, within-subject comparison. It seems likely that the simple action of grasping the soft sock ball and placing it back on the table, often resulted in the target end-state (ball on paper) being achieved inadvertently in the baseline condition. It may also be that the presence of a paper facilitated placing the ball at these locations. The current data does not allow us to distinguish between these accounts. In future applications of the task, the experimenters may consider making the goal-locations less immediately accessible to the child, e. g. by instructing caregivers to place the papers further apart.

Finally, since this study is, to our knowledge, among the first to use online data collection in an imitation paradigm, we would like to end by providing some notes that could help future application of such methods. The present study, and several other developmental psychology studies during the COVID-19 pandemic [e.g., 146], was initially designed for in-lab research and later adapted to online testing. We conducted this study via Zoom due to its availability and popularity during lockdown. Before beginning our primary data collection, we encountered and learned from challenges associated with online testing by piloting. Similar to other online developmental psychology studies [e.g., 146], we observed challenges during the pilot related to environmental distractions in the household, and technical issues related to using the online platform Zoom (e.g., mainly the parents struggling with Zoom). We were able to address these issues during the main data collection for the study by providing Zoom training to the parents during the initial information meeting (Fig 1) and by giving them detailed instructions on how to minimize environmental distractions during the information meeting. For example, they were advised to prevent other family members from entering the room, to tidy up the table used during the test, and to hide any attractive toys within proximity.

As discussed by other developmental researchers [e.g., 147] it might be more difficult to ensure good stimuli presentation in online studies compared to in-lab studies. Indeed, we also found that one limitation of our online procedure was that the researcher could not bring the props to the child herself. We lost data-points due to parents either providing incorrect test objects or accidentally producing the goal-outcome (e.g., ball on paper) when putting the props on the table.

To ensure good quality of presentation of the demonstration videos and to prevent lags that might occur during screen sharing on Zoom, the participating families viewed the videos on their own computer. However, to ensure that the child's web camera captured the child's

attention and their object manipulation on the table, the toddlers often viewed the video stimuli and the live experimenter on a laptop screen that was tilted. Thus, the viewing angle was not optimal, and future online studies could consider ensuring other setups.

On the other hand, one of the major advantages of conducting online testing was the ability to recruit participants from across the country and not just in the city where our lab is located, and thus shortening the period of data collection. Additionally, scheduling multiple appointments on the same day was much easier than in-lab testing, e.g., preparing the experimental set-up was quick with no need for clean up between appointments. Moreover, online testing made it easy to adjust and re-schedule the timing of the appointment to best suit the needs of the child and their family as the family did not have to travel to a lab.

## Conclusion

We sought new type of evidence of toddlers' rich pragmatic inferences about communicative intention. Contrary to our hypothesis, our study did not find evidence that toddlers represent smartphone use as incongruous with ostensive communication and use this representation to infer communicative intention. We indicated how testing of this hypothesis can be improved in the future. A key contribution of this study is the new online imitation paradigm. We have demonstrated that the Sock Ball Task has a potential to become a useful measure of novel subefficient means and goal imitation.

## Supporting information

**S1 Appendix. The faithful imitation score preregistered for Experiment 1.**
(DOCX)

## Acknowledgments

We thank all the families and toddlers who participated in the study. Furthermore, we thank Marianna Osokina and Julia Lea Wilson who helped with coding the data in Experiment 1.

## Author Contributions

**Conceptualization:** Solveig Flatebø, Gabriella Óturai, Mikołaj Hernik.

**Data curation:** Solveig Flatebø.

**Formal analysis:** Solveig Flatebø.

**Investigation:** Solveig Flatebø.

**Methodology:** Solveig Flatebø, Gabriella Óturai, Mikołaj Hernik.

**Project administration:** Solveig Flatebø.

**Resources:** Solveig Flatebø.

**Supervision:** Gabriella Óturai, Mikołaj Hernik.

**Visualization:** Solveig Flatebø, Mikołaj Hernik.

**Writing – original draft:** Solveig Flatebø, Mikołaj Hernik.

**Writing – review & editing:** Solveig Flatebø, Gabriella Óturai, Mikołaj Hernik.

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
