## [Decision Letter · Decision Letter 0]

12 Jan 2024

PONE-D-23-31320No evidence for adult smartphone use affecting attribution of communicative intention in toddlers: online imitation study using the Sock Ball Task.PLOS ONE

Dear Dr. Hernik,

Thank you for submitting your manuscript to PLOS ONE. After careful consideration, we feel that it has merit but does not fully meet PLOS ONE’s publication criteria as it currently stands. Therefore, we invite you to submit a revised version of the manuscript that addresses the points raised during the review process.

The manuscript has been evaluated by two reviewers, and their comments are available below.

The reviewers have raised a number of concerns that need attention. They request additional information on the gaps between related works and this study. 

Could you please revise the manuscript to carefully address the concerns raised?

We look forward to receiving your revised manuscript.

Kind regards,

Laura Kelly

Division Editor

PLOS ONE

Reviewers' comments:

Reviewer's Responses to Questions

**Comments to the Author**

1. Is the manuscript technically sound, and do the data support the conclusions?

Reviewer #1: No

Reviewer #2: Yes

2. Has the statistical analysis been performed appropriately and rigorously? 

Reviewer #1: No

Reviewer #2: Yes

3. Have the authors made all data underlying the findings in their manuscript fully available?

Reviewer #1: No

Reviewer #2: Yes

4. Is the manuscript presented in an intelligible fashion and written in standard English?

Reviewer #1: No

Reviewer #2: Yes

5. Review Comments to the Author

Reviewer #1: 1) Please write the problem statement or research gap ( What you want to do and why) in introduction clearly.

2) Based on these gaps, write your contributions very specifically at the end of Introduction in bullet points.

3) Write the related works group wise. Please make group with similar type of papers and the review those papers analytically. At the end of related works, please write what are you going to solve.

4) You did not write the proposed model clearly. There is no model architecture of you work. How did you solve the problem statement ?

Reviewer #2: The authors investigated the smartphone use and toddler communicative intention attribution. They sought novel type of evidence for toddlers engaging in rich inferences about communicative intentions and specifically, hypothesized that by 18 months, toddlers might be able to recognize one commonly observed category of adult behavior as incongruous with ostension, namely adults’ smartphone use in face-to-face interactions. As a result, toddlers may rely on this interpretation when inferring the communicative intentions of adults.

Strong point:

1. The motivation of this paper is good.

2. The paper is written very well and easy to follow.

3. Experiment results and case studies analysis demonstrates the efficacy of their hypothesis.

Weak points:

1. The participants chosen in the experiments are healthy ones and were born without any complications. How the outcome will vary approximately in case of participants with mixed categories like some were born with mild complications, or not that much healthy etc.?

2. The gaps between existing works and the proposed should be mentioned explicitly.

3. The quality of all the images should be improved.

6. PLOS authors have the option to publish the peer review history of their article (what does this mean?). If published, this will include your full peer review and any attached files.

Reviewer #1: No

Reviewer #2: **Yes: **Md Musfique Anwar

---

## [Author Response · Author response to Decision Letter 0]

25 Jan 2024

For a formatted and easily readable version of this text please see the submitted Letter to the Editor with Response to Reviewers.

Faculty of Health Sciences/

Department of Psychology

Your reference: PONE-D-23-31320

Date:25.01.2024

Laura Kelly

Division Editor

PLOS ONE

Letter to editor with response to Reviewers

Dear Ms Kelly, 

Thank you for your decision letter regarding our submission PONE-D-23-31320 “No evidence for adult smartphone use affecting attribution of communicative intention in toddlers: online imitation study using the Sock Ball Task”. We read carefully both reviews as well as your letter, in which you singled out one point common to both, namely that both reviewers request additional information on the gaps between related works and this study. (For clarity, throughout this letter we use italics to mark yours and the Reviewer’s comments, blue to mark our new text). Thank you for drawing our attention to this issue. Here is how we addressed it:

• We added a new second paragraph in the introduction that ends now with a sentence (underscore added only here for clarity, lines 65-68 in the Revised manuscript, clean copy.):

We will argue that the ubiquitous presence of smartphone use in face-to-face interactions creates a previously unexplored opportunity to further our understanding of toddlers’ early inferences about communicative intentions.

Please note that similar clear statements of novelty of our theoretical framing and hypothesis are already present in the abstract and further in the introduction: “It is unknown when richer pragmatic inferences about communicative intentions emerge in development. We sought novel type of evidence for such inferences in 17-to-19-month-olds.” (abstract, lines 16-18). “But it is still unknown at what point in development children start engaging in this type of rich inferences about communicative intentions. In the current study we sought to answer this question, by investigating 18 –month-olds’ imitation, when they are addressed by a model who uses a smartphone (lines 59-52).

• We added a new paragraph in the introduction (lines 188-197), in which we first summarized the key points from our preceding review of the current literature on phone use in adult-child interactions and its potential impacts, and next we presented how our current study compares to the literature with respect to these key points:

To sum up this short review, the current literature on adult smartphone use in adult-child interactions and its impacts is dominated by observational and parental-survey studies (69-77). The conclusions from the few experimental studies are often severely limited due to the lack of necessary controls (78). One commonly assumed model of parental behavior during smartphone use derived from the still-face paradigm, is likely not capturing the complexity of phenomena that infants are exposed to in real life (64-66). Our approach in the current study was to go beyond these limitations by investigating the potential impact of adult smartphone-use on infant attribution of communicative intention, in an experimental design with carefully matched control. Furthermore, the study was driven by a theoretical proposal that went beyond the current literature reviewed above. We will present it in the next section.

In addition to addressing this key point indicated by you and the Reviewers, we also took great effort to use all the points offered in the two reviews to improve our manuscript. Please see below the detailed list of how we addressed all of the Reviewers’ points. We also used this opportunity to proofread the manuscript. We believe that the text is clearer as a result. We hope that you agree with us on this. Given that none of the Reviewers’ comments indicated any grave issues with our study, we hope that this revised submission will receive your acceptance for publication in PLOS ONE.

Sincerely,

Mikołaj Hernik

Associate Professor

UiT The Arctic University of Norway

–

Mikolaj.l.Hernik@uit.no

+47 77 62 52 59

Reviewer #1: 1) Please write the problem statement or research gap (What you want to do and why) in introduction clearly.

Prompted by the Reviewer’s suggestion we doublechecked that the manuscript indeed contains clear statement of the problem/research gap at the very start of the paper. Here are the relevant places in text:

• (Lines 16-17, third sentence of the Abstract) “It is unknown when richer pragmatic inferences about communicative intentions emerge in development.”

• (Lines 59-60, end of the 1st paragraph of the introduction) “But it is still unknown at what point in development children start engaging in this type of rich inferences about communicative intentions.”

• (Lines 94-96, end of the first review section of the Introduction, now titled “Early sensitivity to communicative intentions”) “But how and when during the first two years of life children enrich their repertoire of pragmatic inferences supporting attribution of communicative intentions is largely unknown.”

2) Based on these gaps, write your contributions very specifically at the end of Introduction in bullet points.

We added now the following new text (in blue), including a new closing paragraph to the introduction (lines 244-258), which spells out very specifically what was done in which experiment.

[…] we hypothesized that children may from early on acquire a representation of smartphone use as incongruous with communicative intention. Furthermore, we expected that representing smartphone use this way may diminish children’s certainty, when attributing communicative intention to an adult who addresses them (i.e., signals intention to communicate) yet also uses a smartphone (i.e., engages in behavior incongruous with communicative intention). […]

In Experiment 1, we tested our general hypothesis, by relying on the assumption that ostensive communication facilitates imitation of novel means in toddlers (94-97). We expected toddlers to imitate less faithfully, if the model, who addressed them ostensively, disrupted the demonstration by engaging in smartphone use, than if she engaged in a matched control behavior (fiddling with a wristwatch). In Experiment 2, we gathered data from two additional conditions: no-demonstration baseline and no-disruption condition, allowing us to assess the validity of our imitation paradigm further.

Note that this new text including the new closing paragraph come at the end of the section of the Introduction now titled (lines 198-199) “From exposure to adult smartphone use to early inferences about communicative intention: the hypothesis”, in which we spell out the argument relating the presented study (specifically Experiment 1) to the problem/knowledge gap mentioned in the Reviewer’s previous point.

3) Write the related works group wise. Please make group with similar type of papers and the review those papers analytically. At the end of related works, please write what are you going to solve.

We added now two new section titles in the introduction to make it clearer that indeed we reviewed papers according to three broad topics. We also separated the final argument-exposition section from the preceding three review-sections. The introduction is now divided into the following sections:

• Early sensitivity to communicative intentions (line 69)

• Smartphone use in face-to-face social interactions

• Smartphone use in face-to-face adult-child interactions

• From exposure to adult smartphone use to early inferences about communicative intention: the hypothesis (lines 198-199)

4) You did not write the proposed model clearly. There is no model architecture of you work. How did you solve the problem statement?

As already mentioned, we gave the final section of the introduction a separate title (From exposure to adult smartphone use to early inferences about communicative intention: the hypothesis) to draw the reader’s attention to the fact that this is where the key argument, hypothesis, and predictions are laid out. This section now ends with new text (see response to the Reviewer’s point 2). As a result, this section gives now full presentation of our argument for how exposure to smartphone use can lead young children to represent smartphone use as incongruous with ostensive communication, leading in turn to uncertainty about communicative intentions of smartphone-users and consequently to predicted lower levels of imitation in our task.

Reviewer #2: The authors investigated the smartphone use and toddler communicative intention attribution. They sought novel type of evidence for toddlers engaging in rich inferences about communicative intentions and specifically, hypothesized that by 18 months, toddlers might be able to recognize one commonly observed category of adult behavior as incongruous with ostension, namely adults’ smartphone use in face-to-face interactions. As a result, toddlers may rely on this interpretation when inferring the communicative intentions of adults.

Strong point:

1. The motivation of this paper is good.

2. The paper is written very well and easy to follow.

3. Experiment results and case studies analysis demonstrates the efficacy of their hypothesis.

We thank the reviewer for recognizing these strengths of our submission.

Weak points:

1. The participants chosen in the experiments are healthy ones and were born without any complications. How the outcome will vary approximately in case of participants with mixed categories like some were born with mild complications, or not that much healthy etc.?

We thank the reviewer for this suggestion, which led us to add the following text to the section of the Discussion titled “Limitations, strengths, and reflections on online testing” (lines 826-833).

A further point to note is that both our general hypothesis and its proposed test through means-imitation task were designed for typically developing children, and it is uncertain whether they may generalize to non-typically developing populations and contexts, for at least two reasons. First, imitation behaviors involving novel actions on objects can vary significantly between typically and non-typically developing children (e.g., 135). Second, children’s prior exposure to adult smartphone use could be influenced by parental factors such as maternal depression (136). Notably in the current study we did not collect any data about the parenting context. 

2. The gaps between existing works and the proposed should be mentioned explicitly.

As already presented in the answer to the Editor above, we paid particular attention to this issue and addressed it by adding new text at the start of the introduction (lines 63-68) and at the end of the section Smartphone use in face-to-face adult-child interactions (lines 188-197).

3. The quality of all the images should be improved.

We doublechecked that all the submitted images are fulfilling the criteria set in PLOS ONE guidelines. However, we used this opportunity to fix small inconsistencies in font-size in Figure 4.

---

## [Decision Letter · Decision Letter 1]

26 Feb 2024

PONE-D-23-31320R1No evidence for adult smartphone use affecting attribution of communicative intention in toddlers: online imitation study using the Sock Ball Task.PLOS ONE

Dear Dr. Hernik,

Thank you for submitting your manuscript to PLOS ONE. After careful consideration, we feel that it has merit but does not fully meet PLOS ONE’s publication criteria as it currently stands. Therefore, we invite you to submit a revised version of the manuscript that addresses the points raised during the review process.

I agree with the reviewers that a good job of editing has been done based on previous comments. despite this a previous reviewer was not available. The paper was then reviewed by a third reviewer, who highlighted the importance of the paper and suggested minor revisions. In particular, the referee highlights an important point regarding the evolutionary phase chosen by the authors. The authors are therefore invited to delve further into the highlighted points.

We look forward to receiving your revised manuscript.

Kind regards,

Giulia Ballarotto

Academic Editor

PLOS ONE

Journal Requirements:

Reviewers' comments:

Reviewer's Responses to Questions

**Comments to the Author**

1. If the authors have adequately addressed your comments raised in a previous round of review and you feel that this manuscript is now acceptable for publication, you may indicate that here to bypass the “Comments to the Author” section, enter your conflict of interest statement in the “Confidential to Editor” section, and submit your "Accept" recommendation.

Reviewer #2: All comments have been addressed

Reviewer #3: (No Response)

2. Is the manuscript technically sound, and do the data support the conclusions?

Reviewer #2: Yes

Reviewer #3: Yes

3. Has the statistical analysis been performed appropriately and rigorously? 

Reviewer #2: N/A

Reviewer #3: Yes

4. Have the authors made all data underlying the findings in their manuscript fully available?

Reviewer #2: Yes

Reviewer #3: Yes

5. Is the manuscript presented in an intelligible fashion and written in standard English?

Reviewer #2: Yes

Reviewer #3: Yes

6. Review Comments to the Author

Reviewer #2: (No Response)

Reviewer #3: Thank you for the opportunity to review the study titled "". The study addresses a central issue in this historical period, allowing for reflection on significant social changes by considering dyadic interaction. I find both studies to be well-structured, and the paper is well-written. However, I believe that the introduction and discussions should further address the competencies and needs of children in this specific age group, as well as the parental role (particularly scaffolding) that could help further support the emerged results. I recommend considering some interesting studies on the topic:

Ballarotto, G., Murray, L., Bozicevic, L., Marzilli, E., Cerniglia, L., Cimino, S., & Tambelli, R. (2023). Parental sensitivity to toddler’s need for autonomy: An empirical study on mother-toddler and father-toddler interactions during feeding and play. Infant Behavior and Development, 73, 101892.

Andreadakis, E., Laurin, J. C., Joussemet, M., & Mageau, G. A. (2020). Toddler temperament, parent stress, and autonomy support. Journal of Child and Family Studies, 29, 3029-3043.

Zhang, H., & Whitebread, D. (2021). Identifying characteristics of parental autonomy support and control in parent–child interactions. Early Child Development and Care, 191(2), 307-320.

Ballarotto, G., Cerniglia, L., Bozicevic, L., Cimino, S., & Tambelli, R. (2021). Mother-child interactions during feeding: A study on maternal sensitivity in dyads with underweight and normal weight toddlers. Appetite, 166, 105438.

Lincoln, C. R., Russell, B. S., Donohue, E. B., & Racine, L. E. (2017). Mother-child interactions and preschoolers’ emotion regulation outcomes: Nurturing autonomous emotion regulation. Journal of Child and Family Studies, 26, 559-573.

7. PLOS authors have the option to publish the peer review history of their article (what does this mean?). If published, this will include your full peer review and any attached files.

Reviewer #2: No

Reviewer #3: No

---

## [Author Response · Author response to Decision Letter 1]

4 Mar 2024

Please note that the same text is available in a more legible, formatted version in the attached Letter to editor with response to reviewer

Dear Dr Ballarotto,

Thank you for your letter and for inviting us to resubmit our manuscript after minor revision. Please see below a quick rundown of how we addressed the reviewer’s feedback:

[Reviewer:]

The study addresses a central issue in this historical period, allowing for reflection on significant social changes by considering dyadic interaction. I find both studies to be well-structured, and the paper is well-written.

[Our response:]

We thank the Reviewer for this positive feedback.

[Reviewer:]

However, I believe that the introduction and discussions should further address the competencies and needs of children in this specific age group, as well as the parental role (particularly scaffolding) that could help further support the emerged results. I recommend considering some interesting studies on the topic:

Ballarotto, G., Murray, L., Bozicevic, L., Marzilli, E., Cerniglia, L., Cimino, S., & Tambelli, R. (2023). Parental sensitivity to toddler’s need for autonomy: An empirical study on mother-toddler and father-toddler interactions during feeding and play. Infant Behavior and Development, 73, 101892.

Andreadakis, E., Laurin, J. C., Joussemet, M., & Mageau, G. A. (2020). Toddler temperament, parent stress, and autonomy support. Journal of Child and Family Studies, 29, 3029-3043.

Zhang, H., & Whitebread, D. (2021). Identifying characteristics of parental autonomy support and control in parent–child interactions. Early Child Development and Care, 191(2), 307-320.

Ballarotto, G., Cerniglia, L., Bozicevic, L., Cimino, S., & Tambelli, R. (2021). Mother-child interactions during feeding: A study on maternal sensitivity in dyads with underweight and normal weight toddlers. Appetite, 166, 105438.

Lincoln, C. R., Russell, B. S., Donohue, E. B., & Racine, L. E. (2017). Mother-child interactions and preschoolers’ emotion regulation outcomes: Nurturing autonomous emotion regulation. Journal of Child and Family Studies, 26, 559-573.

[Our response contd:]

As per Reviewer’s suggestion we addressed the link with the topics and papers raised by the Reviewer both in the Introduction and in the Discussion.

In the Introduction we added a whole short separate paragraph that points to the importance of the topics raised by the Reviewer in connection with our study, while acknowledging that giving them full justice lies beyond the scope of the current paper (lines: 188-193 in the clean manuscipt).

[Our new text:]

Potential impact of parental smartphone use on toddler emotional development and the interplay with factors such as parental stress and support of the child’s autonomy on one hand, and children’s needs, temperament and emotional competencies on the other remain an important topic for future research. However, it lies beyond the scope of the current paper. This broad topic and specific literature (84-88) were brought to our attention by an anonymous reviewer. We come back to it briefly in the discussion.

[Our response contd:]

In the discussion, we added new text where we acknowledge that prior experience with parent scaffolding might have contributed to the lack of predicted effect in toddlers in our study. We thank the Reviewer for pointing us towards this possibility (lines: 765-769 in the clean manuscript)..

[Our new text:]

[…] Relatedly, children’s past experiences with how parents respond to their behavior in emotionally demanding situations, such as parental scaffolding and autonomy support, might foster emotional self-regulation (e.g., ability to wait) when children encounter similar events in the future (85, 88, 124).

[Our response contd:]

We used this opportunity to proofread the manuscript again and corrected minor typos and errors. We hope that with this minor revision the manuscript now finds your acceptance for publication in PLOS ONE.

Sincerely,

Mikolaj Hernik

Associate Professor

mikolaj.l.hernik@uit.no

+47 77 62 52 59

---

## [Editor Report · Decision Letter 2]

6 Mar 2024

No evidence for adult smartphone use affecting attribution of communicative intention in toddlers: online imitation study using the Sock Ball Task.

PONE-D-23-31320R2

Dear Dr. Hernik,

We’re pleased to inform you that your manuscript has been judged scientifically suitable for publication and will be formally accepted for publication once it meets all outstanding technical requirements.

Kind regards,

Giulia Ballarotto

Academic Editor

PLOS ONE
---

## [Editor Report · Acceptance letter]

15 Mar 2024

PONE-D-23-31320R2 

PLOS ONE

Dear Dr. Hernik, 

I'm pleased to inform you that your manuscript has been deemed suitable for publication in PLOS ONE. Congratulations! Your manuscript is now being handed over to our production team.

Kind regards, 

on behalf of

Dr Giulia Ballarotto 

Academic Editor

PLOS ONE